# Relationship between Urine Creatinine and Urine Osmolality in Spot Samples among Men and Women in the Danish Diet Cancer and Health Cohort

**DOI:** 10.3390/toxics9110282

**Published:** 2021-11-01

**Authors:** Selinay Ozdemir, Clara G. Sears, James M. Harrington, Aslak Harbo Poulsen, Jessie Buckley, Chanelle J. Howe, Katherine A. James, Anne Tjonneland, Gregory A. Wellenius, Ole Raaschou-Nielsen, Jaymie Meliker

**Affiliations:** 1Department of Biology, Stony Brook University, Stony Brook, NY 11794, USA; selinayozdemir@gmail.com; 2Department of Epidemiology, Brown University School of Public Health, Providence, RI 02903, USA; clara.sears@louisville.edu (C.G.S.); Chanelle_Howe@brown.edu (C.J.H.); wellenius@bu.edu (G.A.W.); 3Analytical Science Division, RTI International, Research Triangle Park, NC 27709, USA; jharrington@rti.org; 4Danish Cancer Society Research Center, 2100 Copenhagen, Denmark; aslak@cancer.dk (A.H.P.); annet@cancer.dk (A.T.); ole@cancer.dk (O.R.-N.); 5Departments of Environment Health and Engineering & Epidemiology, Johns Hopkins Bloomberg School of Public Health, Baltimore, MD 21205, USA; jbuckl19@jhu.edu; 6Department of Environmental and Occupational Health, Colorado School of Public Health, University of Colorado-Anschutz Medical Campus, Denver, CO 80217, USA; kathy.james@cuanschutz.edu; 7Department of Public Health, University of Copenhagen, 1165 Copenhagen, Denmark; 8Department of Environmental Health, Boston University, Boston, MA 02215, USA; 9Department of Environmental Science, Aarhus University, 4000 Roskilde, Denmark; 10Program in Public Health, Department of Family, Population, & Preventive Medicine, Stony Brook University, Stony Brook, NY 11794, USA

**Keywords:** biomonitoring, urine creatinine, urine dilution, urine normalization, urine osmolality

## Abstract

Assays of urine biomarkers often use urine creatinine to account for urinary dilution, even though creatinine levels are influenced by underlying physiology and muscle catabolism. Urine osmolality—a measure of dissolved particles including ions, glucose, and urea—is thought to provide a more robust marker of urinary dilution but is seldom measured. The relationship between urine osmolality and creatinine is not well understood. We calculated correlation coefficients between urine creatinine and osmolality among 1375 members of a subcohort of the Danish Diet, Cancer, and Health Cohort, and within different subgroups. We used linear regression to relate creatinine with osmolality, and a lasso selection procedure to identify other variables that explain remaining variability in osmolality. Spearman correlation between urine creatinine and osmolality was strong overall (ρ = 0.90; 95% CI: 0.89–0.91) and in most subgroups. Linear regression showed that urine creatinine explained 60% of the variability in urine osmolality, with another 9% explained by urine thallium (Tl), cesium (Cs), and strontium (Sr). Urinary creatinine and osmolality are strongly correlated, although urine Tl, Cs, and Sr might help supplement urine creatinine for purposes of urine dilution adjustment when osmolality is not available.

## 1. Introduction

Urine is often used for non-invasive assessment of exposure to many chemicals, including metals, drugs, nutrients, pollutants, and pesticides, both for research purposes and clinical evaluation. Analyte concentrations in spot urine samples are generally reported using an adjustment to account for variation in dilution across samples.

Urine creatinine is the most commonly used method for standardizing assay results for urinary dilution. Urine creatinine is produced by elimination of serum creatine and creatinine phosphate as a result of muscle metabolic processes [1]. However, because creatinine is associated with muscle mass, which can differ across different segments of the population, concerns have been expressed about its validity as an indicator of urine dilution [1,2,3]. Urine osmolality has been suggested as an alternative, potentially more robust, marker of urine dilution [3]. Urine osmolality provides a measure of dissolved particles, including chloride, glucose, potassium, sodium, and urea in urine [4], and may be a better measure of urine dilution than urine creatinine because osmolality reflects multiple solutes in the urine.

Given the absence of a gold standard for adjusting for urinary dilution, it is presently not possible to objectively determine which measure is better. A couple of studies have evaluated the relationship between creatinine and osmolality in spot urine samples to help clarify their association [3,5]. For example, Yeh and colleagues [3] documented a strong correlation between urine osmolality and urine creatinine in a large sample of US adults (r = 0.75), but also reported a greater influence of socio-demographic and medical conditions on urine creatinine versus osmolality. On the other hand, a small study in HIV patients in Nigeria showed a weaker correlation between urine creatinine and osmolality (r~0.3) [5].

In this study, we investigate the relationship between urine osmolality and urine creatinine using 1375 samples from a subcohort in a population-based adult case-cohort study in Denmark. Our first objective was to quantify the correlation between osmolality and creatinine and examine whether this correlation varies across different subsets of the population defined by age, sex, and disease status. Assuming that osmolality and creatinine are correlated and that osmolality is a preferred yet seldom available marker of urinary dilution, our second objective was to build a prediction model for osmolality that researchers using urine biomarkers in epidemiologic studies could use to predict osmolality based on creatinine and other variables.

## 2. Materials and Methods

### 2.1. Study Population

This study leveraged the Danish Diet, Cancer and Health (DCH) cohort, a longitudinal study that recruited participants and collected samples at baseline between 1993–1997. The DCH cohort consisted of 57,053 persons aged 50–64 at enrollment. The cohort is described in detail elsewhere and was designed to be population-based [6]. Participants, free of cancer at baseline, answered an extensive questionnaire and provided a urinary sample at baseline, stored at −80 °C. The DCH study was accepted by the research ethics committee for Copenhagen and Frederiksberg. Written informed consent was obtained from all participants at enrollment into the cohort.

Our primary analysis used a subcohort from the DCH cohort, comprising 1375 individuals; by design, this included 1200 never smokers and 175 smokers, and 671 women and 704 men. As a secondary analysis, we included all participants from a case-cohort study on stroke, acute myocardial infarction (AMI), and heart failure (HF) nested within the DCH [7,8], which included the randomly selected subcohort (*n* = 1375) and three case populations (AMI: *n* = 985, stroke: *n* = 709, and HF: *n* = 1135); 473 of the identified cases were included in the randomly selected subcohort as expected for a case-cohort design. There were a total of 1740 women and 1991 men in this case-cohort study (Table 1).

At baseline, participants reported sex, age, employment status during the prior year, marital status, smoking status, and secondhand smoke exposure. Participants also self-reported diabetes and completed a dietary questionnaire. Study personnel measured height, weight, and hip and waist circumferences, and we calculated body mass index (BMI) and hip-waist ratio.

We identified incident cases of AMI, HF, stroke, and diabetes that occurred between baseline and 31 December 2015 (AMI and HF), 31 December 2012 (diabetes), or 30 November 2009 (stroke) using ICD codes recorded in the Danish National Patient Registry and the Danish National Diabetes Registry. 

### 2.2. Urine Analyses

A 1 mL urine sample from each participant was analyzed at RTI International’s Trace Metals Laboratory (Research Triangle Park, NC, USA). We allowed the samples to thaw to room temperature and then resuspended any precipitate or particles by rotating the samples end over end (gently to prevent bubbling) for at least 60 s per sample. Urinary creatinine was quantified colorimetrically by the Jaffe reaction with a Cayman Chemicals (Ann Arbor, MI, USA) Creatinine Assay Kit. Recovery of the standard reference material (SRM) generally fell within 90–110%. Approximately 10% of samples were reanalyzed as incurred samples after completion of all measurements, and the coefficient of variation (CV) = 3.9% across those incurred samples. Urinary osmolality was measured by a Model 3320 Micro-Osmometer by Advanced Instruments, Inc. (Norwood, MA, USA). Clinitrol Reference Solution (Advanced Instruments, certified value 290 mOsm) and 800 mOsm Renol Urine Osmolality Control solution were used to verify instrument performance daily, and we reported CV of 1.3%, 0.7%, respectively, for the two reference solutions. All measured values fell within 5% of the expected value.

An iCAP Q ICP-MS system (Thermo Scientific, Waltham, MA, USA) equipped with a collision cell was used for the determination of a suite of 19 elements as described previously (Poulsen et al., 2021, Sears et al., 2021). Urinary cotinine was measured by a cotinine ELISA bioassay kit by Abnova Corporation (Taipei, Taiwan).

### 2.3. Statistical Analyses

Our primary analysis was restricted to the subcohort, which should be generalizable to the full population-based cohort. Results for the case-cohort sample are shown in the Appendix A; even though the case-cohort sample was expected to not be representative of the parent cohort, it increased the available sample size and statistical power for stratified analyses. All analyses were conducted using SAS V9.4 (Cary, NC, USA).

We first created Bland-Altman plots using z-score standardized creatinine and osmolality to compare the mean of creatinine and osmolality with the difference between creatinine and osmolality. This figure provided a visualization of the extent to which the differences between creatinine and osmolality varied across the range of values (Appendix A).

We next calculated correlation coefficients between urine creatinine and urine osmolality. Creatinine followed a log-normal distribution, so we calculated Spearman coefficients, and, for comparison, we also calculated Pearson coefficients. We calculated correlation coefficients in the subcohort and in different strata of the subcohort defined by: sex; never versus current smoking; BMI 15–25 kg/m^2^, BMI 25–30 kg/m^2^, BMI ≥ 30 kg/m^2^; experience of heart failure, AMI, and stroke; and experience of diabetes separately using self-reports at baseline and incidence reports from the registry.

We next sought to develop a prediction model for osmolality based on urine creatinine, key patient characteristics, and other common analytes. We first modeled osmolality (dependent variable) in relation to creatinine alone (Model 1). Next, we used a lasso variable selection procedure (Flom and Cassell 2007) to identify which of the following commonly available variables improved the model beyond the inclusion of creatinine alone (Model 2): age in years, employment status, marital status, smoking status, height, sitting height, hip-waist ratio, body weight, BMI, self-reported disease status at baseline (diabetes, high blood pressure, high cholesterol), disease status subsequent to baseline via registry linkage (HF, AMI, stroke, and diabetes), baseline non-steroidal anti-inflammatory drugs (NSAIDs), baseline aspirin, history of hormone replacement therapy, and oral contraceptive history. Model 3 continued to use the lasso procedure and included the variables considered in Model 2 plus less commonly available variables: secondhand smoke exposure, urine cotinine (µg/L), dietary intake from food frequency questionnaire and dietary calculations (fish, red meat, vegetables, fruit, fat, saturated fat, refined sugars, alcohol, calories), estimated nutrient intake from food and supplements (Zn, K, Ca, Fe, M), and urine metals (Co, Zn, As, Se, Sr, Mo, Cd, Sn, Sb, Cs, Ba, Hg, Pb). Variables were parameterized continuously where possible, with the exception of BMI, which was coded 15–25, 25–30, and ≥30 with 25–30 as the reference group. We also considered up to 7th degree polynomials for all continuous variables.

As a sensitivity analysis, we also used a least angle regression (LAR) selection procedure, and results were fully consistent with the lasso approach and therefore are not reported. In another sensitivity analysis, we included only participants with creatinine values between 129 and 2690 μg/L (women) and 204–3520 (men), which are considered the normal ranges for 95% of the population [7]; this reduced the sample size from 1375 to 1298 in the subcohort.

## 3. Results

Descriptive characteristics in the subcohort and in different segments of the subcohort are shown in Table 1. Subsets with lower levels of osmolality tended to have lower levels of creatinine. Creatinine and osmolality were both lower among women, older participants, and among those with BMI 15–25. Urine creatinine and osmolality levels were similar between the subcohort and the case-cohort study population (Appendix A).

Figure 1 and Appendix A report correlation coefficients for the association between urine creatinine and osmolality in the subcohort and different subsets of the subcohort. Correlation between creatine and osmolality was strong overall (Spearman = 0.90; Pearson = 0.82) and similarly strong for most subsets of the subcohort. The correlation was weaker for individuals with diabetes, especially those who had diabetes at cohort baseline, for those who were obese (BMI > 30), and also for HF and stroke case populations. The correlation was somewhat stronger for those with BMI 15–25 compared with BMI > 30. Correlation coefficients for the case-cohort study sample are also reported in Appendix A and indicate similar trends. Bland-Altman plots suggested a strong correlation but also revealed greater variability at higher levels of urine creatinine and urine osmolality (Appendix A).

The linear regression analysis showed that 60% of the variability of urine osmolality was explained by urine creatinine alone (Model 1; Table 2). None of the commonly available covariates, i.e., demographics, medications, disease status, BMI, or smoking status were selected by the lasso procedure as being meaningfully associated with osmolality (Model 2). When we included the less commonly available covariates, i.e., dietary and urinary markers, some urine metals were selected by the lasso procedure as being associated with osmolality, including strontium (Sr), cesium (Cs), and thallium (Tl), with an Adj R^2^ = 0.69. In the sensitivity analysis including only those samples within the normal range for urine creatinine, the same parameters were identified with highly similar magnitudes of associations, with an overall increase in model fit to Adj R^2^ = 0.75 and MSE = 17,203. In the case-cohort study sample, results were highly similar (Appendix A).

## 4. Discussion

Adjusting for variation in urine dilution is critical in understanding urine analyte concentrations in spot urine samples. Creatinine is most commonly used to account for urine dilution, although some have suggested that osmolality may be an alternative, potentially more robust, marker of urine dilution [3]. The degree to which urine creatinine and osmolality are correlated is not well understood, especially whether there are differences across strata defined by age, sex, BMI, or other factors. This study indicates a strong correlation (Spearman’s ρ = 0.90, Pearson’s r = 0.82) between urine creatinine and urine osmolality in adult men and women in the randomly selected subcohort of the population-based Danish Diet, Cancer, and Health Cohort. The relationship was similarly strong in the case-cohort study population and in different segments of the subcohort and case-cohort populations, although slightly weaker correlations were seen among participants who had diabetes at baseline or who developed diabetes after urine collection, and those with BMI > 30. Regression models predicting osmolality showed that creatinine explained 60% of the variability in osmolality. More commonly available covariates did not improve model fit; however, urine measures of Tl, Cs, and Sr explained another 9% of the variance.

One previous study in a large sample of NHANES also reported a strong correlation between urine osmolality and urine creatinine (Pearson’s r = 0.75) but did not investigate the correlation in subsets of the population [3]. However, they reported that diabetes and chronic kidney disease (CKD) had greater impacts on creatinine levels, and protein intake had a greater impact on osmolality levels. Another study reported on the ratio between urine osmolality and urine creatinine across different age groups and concluded that the ratio did not vary by any of the factors considered, which included age, sex, body weight, or height [8]. Collectively, these studies and ours suggest that most factors do not strongly impact the relationship between urine creatinine and urine osmolality, indicating they may be interchangeable as markers of urine dilution, with the exceptions being in participants with diabetes, CKD, and high protein intake. For clinical significance, this indicates that either osmolality or creatinine can be used to control for urine dilution in most subsets of the population, although clinicians should consider the health of the kidneys and protein intake patterns when interpreting urinary biomarkers.

Our study population did not include those with CKD, but we did include individuals with diabetes, and we had rich dietary data from a food frequency questionnaire. Neither these variables nor other demographic, disease, or dietary variables explained additional variance in the regression models. Urine Tl, Cs, and Sr explained some of the variance in osmolality in the regression model. Sr is an essential micronutrient; diet is a common route of exposure and leads to accumulation in the bone and excretion in the urine and feces [9]. Levels of Sr in the urine typically reflect dietary exposure in addition to normal bone remodeling [10]. Tl is a known neurotoxin, and levels in urine are correlated with levels of exposure, in which urine levels can stay elevated for at least a year following heightened exposure [11,12]. Retention time in the excretion of Cs is related to muscle mass [13,14,15], and Cs levels are slightly higher in the skeletal muscle than other tissue. Excretion of urine Cs may be related to muscle catabolism; therefore, factoring it into a model may help mitigate the influence of muscle catabolism on urine dilution adjustment with creatinine. We are not sure why Sr and Tl in urine might help predict urine osmolality above and beyond the association with creatinine, but investigators might also wish to consider exploring the addition of these measures along with creatinine when controlling for urine dilution. 

A few recent studies compared the impact of different methods of dilution adjustment on exposure estimates or disease outcomes [16,17,18,19,20]. One study showed no difference in associations between creatinine adjustment and osmolality adjustment using metabolomic data in a small study of 51 samples [16]. In our case-cohort studies, we did not observe strong differences in hazard ratio estimates after adjustment for creatine versus osmolality [19,20]. Two other studies showed osmolality adjustment to be superior to creatinine adjustment in associations with predictors of exposure or urinary flow rate [17,18], in line with the presupposition that osmolality may be less susceptible to underlying physiology or pathophysiology, such as differences in muscle catabolism, in comparison with creatinine. Nonetheless, both measures appear to be influenced by BMI, sex, and other covariates (results not shown), suggesting that covariate-adjusted standardization, which has been proposed for creatinine [21], may also be useful when correcting urinary biomarker concentrations for osmolality.

Some advantages of our study include the large sample size, the rigorous laboratory analysis of creatinine, osmolality, and trace metals, and the availability of complementary demographic and dietary data that can support our interpretation of the relationship between urine creatinine and osmolality. Even though we adopted a lasso approach, which is a more robust variable selection procedure compared with forward selection, any automatic variable selection procedure may find nonsensical associations [22]. However, we attempted to only include variables that plausibly made sense; therefore, concerns about nonsensical associations were likely limited in our analysis. We also confirmed that the same variables were identified using the LAR approach, but that does not fully mitigate this limitation. Our study was also limited in its ability to clarify which measure is best for controlling for urine dilution because we do not have a gold standard. In addition, our results may only be applicable to predominantly white, non-Hispanic participants aged 50–65, although our urine measures were in the range expected for the general population [2,3,23,24]. Further, we did not have complete data on chronic kidney disease, subclinical kidney damage, or those taking diuretics, but prevalence should be low in this population, as expected for a population-representative cohort aged 50–65.

This large study indicated that creatinine and osmolality are highly correlated, but still, ~40% of the variability in osmolality was not explained by creatinine. Adding urine measures of the elements Cs, Tl, and Sr helped to explain an additional 9% of the variability. Urine osmolality may, theoretically, be a preferred measure because it comprises multiple solutes and is not influenced by muscle catabolism. Including measures of urine Cs, Tl, and Sr, in addition to urine creatinine, may prove beneficial for urine dilution adjustment in human studies in which osmolality is not available, although this should be verified in a follow-up study in a more diverse cohort.

## Figures and Tables

**Figure 1 toxics-09-00282-f001:**
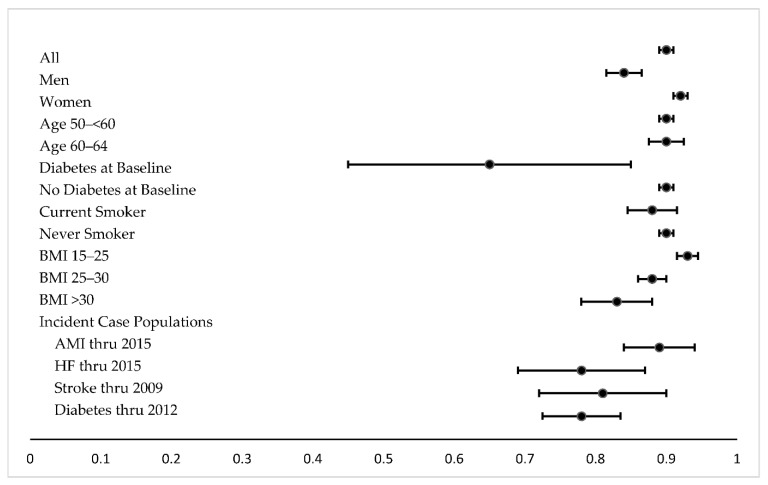
Spearman correlation coefficients in subcohort. 95% CIs are depicted, using Fisher’s Z transformation.

**Table 1 toxics-09-00282-t001:** Characteristics of the Subcohort.

	N (%)	Median, 25th–75th %iles: Cr, mg/L	Median, 25th–75th %iles: Osmolality, mOsm
All	1375	995 (462–1650)	585 (313–784)
Men	704 (51)	1330 (777–1910)	693 (443–848)
Women	671 (49)	654 (311–1275)	448 (237–707)
Age 50–<60	1045 (76)	1030 (485–1710)	603 (322–795)
Age 60–64	330 (24)	860 (399–1440)	533 (280–735)
Diabetes at baseline	19 (1)	1130 (655–1800)	762 (609–901)
Current Smoker	175 (13)	1040 (461–1920)	610 (322–789)
Never Smoker	1200 (87)	987 (466–1640)	583 (312–784)
BMI 15–25	562 (41)	871 (379–1440)	509 (266–746)
BMI 25–30	587 (43)	1070 (515–1730)	623 (327–797)
BMI ≥ 30	226 (16)	1225 (586–1790)	689 (448–849)
Incident Case Populations			
AMI thru 2015	62 (5)	1130 (595–1710)	632 (355–802)
HF thru 2015	64 (5)	1170 (461–1880)	659 (327–816)
Stroke thru 2009	47 (3)	1100 (659–1870)	659 (492–824)
Diabetes thru 2012	201 (15)	1210 (586–1870)	704 (431–824)

CR = creatinine; BMI = body mass index; HF = heart failure; AMI = acute myocardial infarction.

**Table 2 toxics-09-00282-t002:** Model R^2^, mean square error, and β coefficients of predictor variables selected in lasso procedure in relation to urine osmolality (mOsm).

	β Coefficient ^#^	Mean Square Error	R^2^
Model 1 and Model 2 *		26,685	0.60
Urine Creatinine (mg/L)	0.27		
Model 3		20,332	0.69
Urine Creatinine (mg/L)	0.19		
Urine Strontium (µg/L)	0.20		
Urine Cesium (µg/L)	10.21		
Urine Thallium (µg/L)	279.94		

* Results from Model 2 only showed creatinine associated with osmolality. **^#^** These β coefficients indicate the change in osmolality per 1 unit increase in the predictor variable. The middle 50th percentile values for Sr were 116–336 µg/L, for Cs were 2.38–6.66 µg/L, and for Tl were 0.10–0.27 µg/L.

## Data Availability

Data are available from the Diet, Cancer and Health Institutional Data Access (https://www.cancer.dk/research/diet-genes-environment/dgedch/); data were first accessed for this grant funded project in 2017. The application form along with a list of variables can be obtained from Louise Hansen (kd.recnac@nahuol). To access data, an application must be approved by the Scientific Board. Furthermore, as data contain potentially identifying or sensitive information, access to data has to be registered and approved by The Danish Data Protection Agency (https://www.datatilsynet.dk/english/) and/or a Health Research Ethics Committee (http://www.nvk.dk/english).

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
