# Peer review of "Relationship between Urine Creatinine and Urine Osmolality in Spot Samples among Men and Women in the Danish Diet Cancer and Health Cohort"

_toxics, 2021, doi:10.3390/toxics9110282_

Round 1
Reviewer 1 Report
The development of a urine dilution correction method to replace creatinine or to complement the weaknesses of creatinine is needed, and we believe that this study is highly important.
Please consider the following
Methods
#1. In frozen urines, urinary components precipitate and crystallize. If crystals precipitate in urine, osmolality cannot be measured correctly. How did you avoid this in this study ?
Results
#1. The unit of osmolality is listed as mOsm, but does it mean mOsm/L?
#2. Please show the results of trace element analysis (Median, 25-75%tile).
#3. It may be due to my lack of understanding, but I do not understand how to read Fig.1. The item names (All, Men, etc.) do not match the number of bars in the correlation coefficient.
Author Response
Point-by-point response is provided on the attached. Thank you.

Reviewer 2 Report
The research was planned and carried out correctly and the obtained results were discussed very well.
I believe that despite the analytical determinations have been carried out in a reputable institution, the basic validation parameters should be given anyway, it is not absolutely necessary
Author Response

(The authors gave the same response as above.)
